# Selective retention of virus-specific tissue-resident T cells in healed skin after recovery from herpes zoster

Kerry J. Laing [1][✉], Werner J. D. Ouwendijk [2], Victoria L. Campbell [1], Christopher L. McClurkan[1], Shahin Mortazavi[1], Michael Elder Waters[1], Maxwell P. Krist[1], Richard Tu[1], Nhi Nguyen[1], Krithi Basu[1], Congrong Miao[3], D. Scott Schmid[3], Christine Johnston[1,4], Georges M. G. M. Verjans [2] & David M. Koelle[1,4,5,6,7]

Herpes zoster is a localized skin infection caused by reactivation of latent varicella-zoster virus. Tissue-resident T cells likely control skin infections. Zoster provides a unique opportunity to determine if focal reinfection of human skin boosts local or disseminated antigen-specific tissue-resident T cells. Here, we show virus-specific T cells are retained over one year in serial samples of rash site and contralateral unaffected skin of individuals recovered from zoster. Consistent with zoster resolution, viral DNA is largely undetectable on skin from day 90 and virus-specific B and T cells decline in blood. In skin, there is selective infiltration and long-term persistence of varicella-zoster virus-specific T cells in the rash site relative to the contralateral site. The skin T cell infiltrates express the canonical tissue-resident T cell markers CD69 and CD103. These findings show that zoster promotes spatially-restricted long-term retention of antigen-specific tissue-resident T cells in previously infected skin.

Tissue-resident-memory T cells ($T_{RM}$) are a phenotypically distinct subset of memory T cells confined to tissues where they may encounter their cognate antigen, permitting a rapid, local, and presumably beneficial immune response. Though locally mobile, $T_{RM}$ are largely non-migratory with regards to exit to the circulation[1,2]. Originating as early effector T cells[3], local inflammation and T cell receptor (TCR) signaling may program the long-lived $T_{RM}$ phenotype[4], which includes a conserved set of tissue retention molecules, such as CD69 and to some extent CD103 and CD49a, and location-dependent transcription factors, such as Notch, Runx3, Hobit, and Blimp-1[5]. $T_{RM}$ in skin are likely important in protection against skin-tropic infectious agents.

Herpes zoster (HZ), also known as shingles, is a painful, vesicular skin eruption caused by the reactivation of varicella-zoster virus (VZV) from ganglionic neurons that remain chronically infected after primary varicella. Pain lasting after HZ often reduces quality of life[6] while eye or cranial blood vessel involvement can lead to severe complications such as blindness or stroke[6]. Unlike varicella, which presents as a disseminated skin infection, HZ typically affects a single dermatome. The mechanisms of viral clearance by adaptive immunity in either case are not fully elucidated. Antibodies limit primary infection[7] and reduce VZV intercellular spread[8] and could be an effector mechanism during HZ. CD4 and CD8 T cells, however, are considered more important for minimizing HZ[8,9] and limiting disease severity, as reflected by the

[1]Department of Medicine, University of Washington, Seattle, WA, USA. [2]HerpeslabNL of the Department of Viroscience, Erasmus Medical Center, Rotterdam, The Netherlands. [3]Centers for Disease Control and Prevention, Division of Viral Diseases, Atlanta, GA, USA. [4]Vaccine and Infectious Diseases Division, Fred Hutchinson Cancer Center, Seattle, WA, USA. [5]Department of Laboratory Medicine and Pathology, University of Washington, Seattle, WA, USA. [6]Department of Global Health, University of Washington, Seattle, WA, USA. [7]Department of Translational Research, Benaroya Research Institute, Seattle, WA, USA. [✉]e-mail: laingk@uw.edu

profoundly increased incidence of HZ in persons with advanced HIV-AIDS[10].

In persons with HZ, systemic VZV-specific T cells are elevated[11] and reactivating ganglia and HZ rashes become intensely inflamed with T cell infiltrates[12]. T cell recruitment and enrichment in VZV-infected skin may mirror the well-documented herpes simplex virus (HSV) phenotype, since the pathogeneses of recurrent infection by these related neuro- and skin-tropic alphaherpesviruses are similar[13,14]. Yet, there are major temporal differences in HSV and VZV recurrences. Ganglionic reactivation and axonal delivery of HSV-2 to skin occurs repeatedly at the same sites, leading to frequent, short (hours to days) bursts of detectable virus[15]. This provides intermittent boosts of inflammatory stimuli and HSV antigen that maintain local HSV-2-specific T cells in skin and reproductive tract between recurrences, even during chronic antiviral therapy[16,17]. In contrast, VZV lingers up to several weeks following HZ[18] before clearance, leaving a healing skin site devoid of viral antigen. HZ recurrences are uncommon in immunocompetent persons and seldom occur twice in the same dermatome[19], suggesting local, site-specific protective immunity. Animal models[20] are not amenable to study local retention of VZV-specific $T_{RM}$. Mouse models of local skin infection by other skin-tropic viruses, HSV-1 and vaccinia, show disparate anatomical localization of virus-specific $T_{RM}$ after viral clearance. HSV-1 systems that lack epithelial reactivation show profoundly localized long-term retention of $T_{RM}$ with prompt antiviral function[21]. In contrast, local skin vaccinia infection leads to disseminated skin CD8 $T_{RM}$ in mice[2].

Here, we use HZ as a physiological probe to test the hypothesis that a single, anatomically restricted, intense exposure to viral antigen leads to localized deposition of long-lasting virus-specific $T_{RM}$ in human skin.

## Results

### Subjects and specimens

We recruited 19 persons with a clinical diagnosis of HZ. One was adjudicated to not have HZ based on the rash characteristics and was excluded. Amongst 18 persons with HZ (Supplementary Table 1), 10 were male and 8 were female, two of each were Black or Asian, one was multiracial, and 13 were white. Since age is an important determinant of HZ risk, we included subjects across a wide range of ages: 3 subjects were under 30, 2 subjects were 30–39, 6 subjects were 40–49, 3 subjects were 50–59, and 4 subjects were over 60 years old. The median age was 47 years (range 21–71). HLA types were diverse; three persons had some degree of HLA-A or B homozygosity. None had immune suppressive conditions or chronic medications. Most HZ rashes involved the thoracic dermatome ($N$ = 13, 72%). Amongst 17 persons receiving oral antiviral therapy, 2 also received short-course oral corticosteroids. Three subjects self-reported receipt of a zoster vaccine: subject 81 (recombinant zoster vaccine, 48 months prior to HZ), subject 89 (unknown type and time), subject 91 (recombinant zoster vaccine, unknown time). Varicella vaccination status was not captured but most subjects were born before universal pediatric varicella vaccination began in the US in 1995. One person (subject 89, Supplementary Table 1) declined all biopsies, 6 subjects declined biopsies after day 90, and 11 completed the protocol to one year. One subject (subject 80) had an additional day 500 biopsy.

### Virus and systemic immunity dynamics

HZ and contralateral (CON) skin swabs were obtained before biopsy and analyzed for VZV DNA by real-time PCR (Fig. 1A, Supplementary Table 2). One month after rash onset, VZV DNA was detected in 13 of 18 (72%) persons at the rash site with median VZV DNA viral load of 1513 DNA copies/swab (95% confidence interval of median (CI) 0–11,250). VZV DNA detection dropped significantly by day 45 (8/17 swabs positive, median 0, 95% CI 0–1,932; $p$ = 0.016 vs. day 30) and day 90 (2/17 swabs positive, median 0, 95% CI 0–0; $p$ = 0.0078 vs. day 45). No subjects had detectable VZV DNA at 1 year. Low level VZV DNA was

detected once on CON skin (257 VZV DNA copies/swab; day 360, subject 78). All specimens were negative for HSV DNA.

Systemic VZV-specific CD4 T cell and humoral responses declined over time after HZ (Fig. 1, Supplementary Table 2). The median VZV-specific CD4 T cell frequency after HZ—defined as the net (mock subtracted) proportion of cells accumulating IFN-γ and/or IL-2 after VZV antigen exposure—was 0.15% (95% CI 0.07–0.22) of CD4 T cells at day 45, a 53% drop to 0.07% (95% CI 0.04–0.13, $p$ = 0.0047) at day 90, and remaining at 0.06% (95% CI 0.02–0.12; $p$ = 0.58 vs. day 90) at one year (Fig. 1B). The titer of VZV glycoprotein-specific IgG, represented by $OD_{450}$ values from ELISA using a dilution of 1:1280, declined by 17% from a median 2.03 (95% CI 1.09–3.08) at day 45 to 1.69 (95% CI 1.34–2.33, $p$ < 0.0001 vs. day 45) by day 60, to 1.57 (95% CI 0.92–1.89, 7.1% decrease vs. day 60, $p$ = 0.0003) by day 90, and to 1.02 by day 360 (95% CI 0.49–1.72, 35% decrease vs. day 90, $p$ = 0.001) (Fig. 1C, Supplementary Table 3). Similarly, neutralizing antibody titers, measured as complement-dependent microneutralization, declined over time from a median titer of 640 at day 45 (95% CI 320–1,280), day 60 (95% CI 320–640), and day 90 (95% CI 320–640) to a titer of 160 at day 360 (95% CI 160–640; 75% decrease vs. day 90, $p$ = 0.002) (Fig. 1D). The decline in systemic responses after clinical resolution matches the kinetics of serial VZV DNA PCR results, and suggests that systemic immunity declines after antigen clearance.

### In situ analysis of T cells in skin after herpes zoster

To determine the abundance, phenotype and location of T cells in skin after HZ we analyzed cryosections by immunofluorescence. Despite VZV DNA presence in some swabs (Fig. 1A), VZV immediate early protein IE63 (a marker for productive VZV infection) was not detected in any biopsy. A transient (71%) increase in T cells (HZ median 221 cells/mm², 95% CI 9.4–322.3; CON median 64 cells/mm², 95% CI 8.2–185.3, $p$ = 0.009), especially CD8 T cells, was observed in HZ compared to CON skin (Fig. 2A, C, Supplementary Table 4). T cells were predominantly located in clusters in the dermis close to hair follicles and/or vasculature, with lower numbers in the epidermis. Most T cells expressed the canonical $T_{RM}$ marker CD69, a subset of which co-expressed CD103, especially among epidermal T cells (Fig. 2B, D, Supplementary Table 4). An increase in T cells expressing the cytolytic granule marker TIA-1 was observed in healed HZ skin at day 45 (Fig. 2E, G, Supplementary Table 4), while a subset of CD8-negative T cells located in dermal infiltrates at days 90 and 360 post-HZ expressed regulatory T cell marker FOXP3 (Fig. 2E, F, Supplementary Table 4). Collectively, the data indicate that HZ is associated with a selective, transient increase in T cells expressing $T_{RM}$ markers in the affected skin. By day 360, the overall density of CD4 and CD8 T cells appeared to return to levels comparable to paired contralateral skin.

### VZV-specific CD4 T cells selectively increase in HZ skin

Skin-derived T cell lines (skin-TCL) comprising both CD4 and CD8 T cells were assessed for VZV-specificity. First, we measured CD4 T cell responses to whole VZV antigen using ICS (representative example, Fig. 3A). Measures in skin-TCL were considered a semi-quantitative estimate of skin-resident VZV-specific CD4 T cells. HZ skin-derived TCL (HZ-TCL) were often rich (up to 73% of CD4 T cells) in VZV-reactive CD4 T cells whereas frequencies in contralateral-TCL (CON-TCL) were much lower. Significantly higher frequencies of VZV-specific CD4 T cells were found at each time point in HZ-TCL (day 45, median 1.1% [95% CI 0.1–7.0, $n$ = 17], day 90 median 3.6% [95% CI 0.1–14.9, $n$ = 17], day 360 median 0.1% [95% CI 0–4.1, $n$ = 11]) compared to CON-TCL (day 45, median 0.02% [95% CI 0–0.2, $n$ = 17, $p$ = 0.006], day 90 median 0% [95% CI 0–0.14, $n$ = 17, $p$ = 0.0006], day 360 median 0% [95% CI 0–0, $n$ = 11, $p$ = 0.03]) (Fig. 3B). Levels of >3% VZV-specific CD4 T cells were suitable to determine which viral antigens[22] were recognized when autologous PBMC were used as antigen presenting cells (APC)[23–26] (Supplementary Fig. 1A). HZ-TCL derived from VZV DNA-negative skin

frequently had antigen-specific responses, consistent with a $T_{RM}$ phenotype (Supplementary Table 2). Strong, discrete responses to a number of VZV proteins were noted (representative data, Supplemental Fig. 1B, C). For HZ-TCL from day 45, up to 10 VZV proteins (median 2, 95% CI, 0–10.0, $n = 8$) invoked proliferation (Fig. 3C). Antigen breadth was similar at day 90 (median 1, 95% CI 0–3; $n = 9$). Two HZ-TCL from day 360 had proliferative responses. For CON-TCL fewer proliferative responses were observed at day 45 (median 0.5, 95% CI, 0–3, $n = 8$; $p = 0.047$). CD4 T cell antigenicity was noted to 31/71 (44%) VZV proteins overall (Fig. 3D) and with highest prevalence to VZV glycoproteins gB (ORF31), gI (ORF67), and gE (ORF68), mirroring responses in blood[26]. These assays demonstrate selective retention of VZV antigen-specific CD4 $T_{RM}$ in HZ skin compared to unaffected skin that remain locally detectable for up to one year in the absence of cognate antigen.

## VZV-specific CD8 T cells selectively increase in HZ skin

To detect and determine the breadth of VZV-specific CD8 T cells in skin TCL, we used an artificial APC (aAPC) platform of COS-7 cells co-transfected with individual VZV ORFs and subject-specific HLA-A or B alleles. Strong CD8 T cell responses to discrete HLA/VZV ORF combinations were detected using the criteria described in Methods (representative screens, Supplementary Fig. 2). Among 173 HLA-level proteome-wide screens of HZ-TCL from 17 HZ subjects, 67 (39%) were

positive for at least one VZV ORF (Fig. 4A). Almost all HLA alleles studied showed reactivity in at least one HZ-TCL, indicating widespread antigenicity in HLA-diverse subjects. Of note, VZV ORF9 was highly immunoprevalent in skin-derived TCL (14/17 [82%] subjects Fig. 4B]) and also blood-derived TCL, derived from a separate cohort of 10 healthy persons (Supplementary Fig. 3). In all, responses represented 15 HLA class I allelic variants covering a majority of the human population[27].

Overall, VZV-specific CD8 T cell reactivity was detected in HZ-TCL of 15/17 (88%) subjects and were often readily detectable in day 360 HZ-TCL. In contrast, CON-TCL rarely showed VZV-specific CD8 T cell responses. CD8 T cell antigenic breadth decreased in HZ-TCL from 4 VZV proteins (median, 95% CI 2–4, $n = 17$) at day 45 to 2 proteins by day 90 (95% CI 1–2, $n = 17$) and one protein by day 360 (95% CI 0–3; $n = 11$). The number of ORFs recognized by CD8 T cells in HZ-TCL was significantly higher than the CON-TCL at both day 45 (median 0, 95% CI 0–1, $p = 0.0007$) and day 90 (median 0, 95% CI 0-1, $p = 0.009$) (Fig. 4C). These data show that VZV-specific CD8 T cells selectively infiltrate HZ skin and are retained as $T_{RM}$ for at least 90 days and for up to 1 year after resolution of HZ skin rash.

## Quantification of VZV epitope-specific CD8 T cells in skin

T cell epitope mapping with synthetic peptides and aAPC confirmed antigen specificity and HLA restriction (representative CD8 T cell

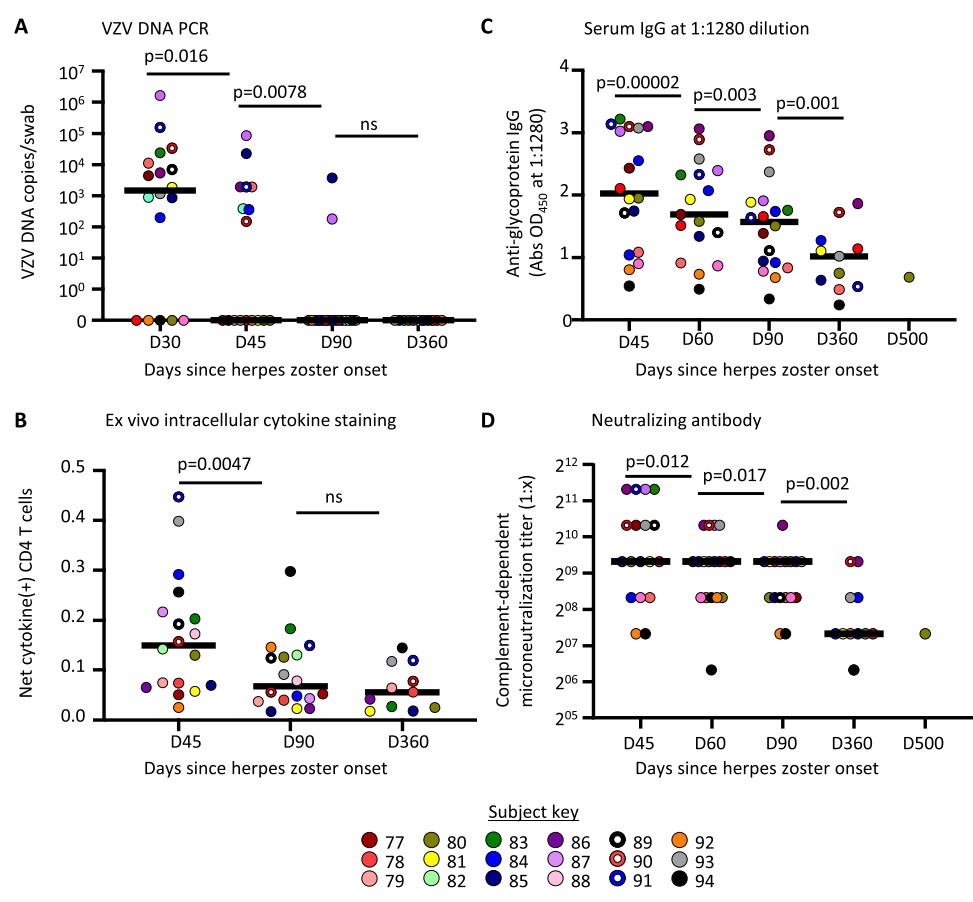

**Fig. 1 | Varicella-zoster virus (VZV) clearance and systemic immunity after herpes zoster (HZ). A** VZV DNA copy number was determined by polymerase chain reaction (PCR) from swabs taken from the HZ rash site at the indicated time frames (D = day) after HZ onset. **B** Net frequency of CD4 T cells in PBMC with VZV-specific interferon-γ (IFN-γ) and/or interleukin 2 (IL-2) responses. **C** Serum IgG responses to mixed VZV glycoproteins measured by ELISA at a serum dilution of 1:1280. **D** Complement-dependent VZV microneutralization titer of sera over time.

Panels show conserved color dots representing each study subject and the median (solid line) levels of analyte within each time point. Unadjusted two-tailed $p$ values from multiple Wilcoxon tests to compare sequential time points are shown. Number of subjects tested per time point: D30, $n = 18$ (**A**); D45, $n = 17$ (**A–D**); D60, $n = 17$ (**C**, **D**); D90, $n = 17$ (**A–D**); D360, $n = 11$ (**A–D**), D500 (**C**, **D**). Abs OD$_{450}$, absorbance optical density at 450 nanometers. Source data are provided as a Source Data file.

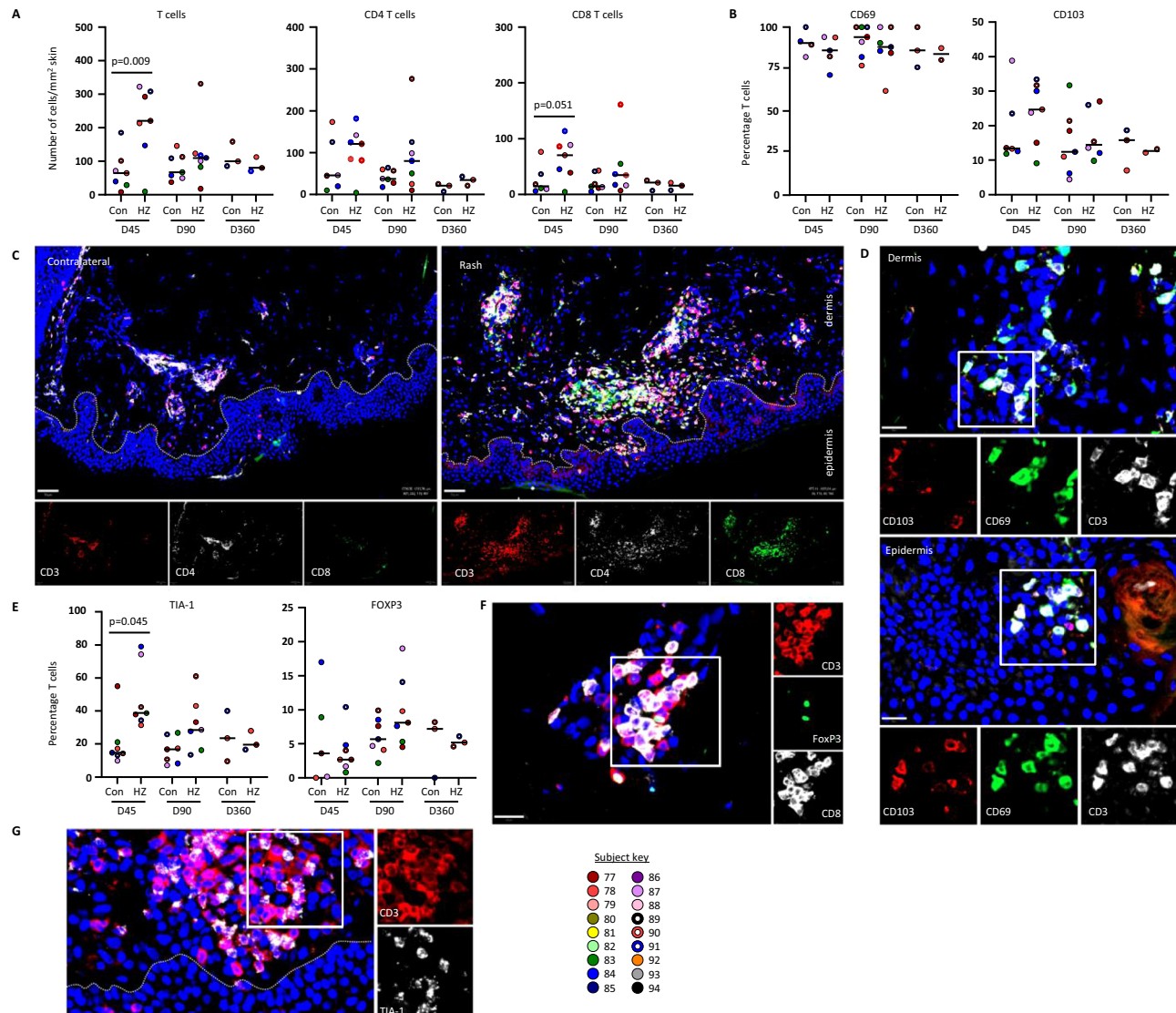

**Fig. 2 | Detection T_RM cells in herpes zoster-affected and contralateral human skin biopsies.** Cryosections from rash site (HZ) and contralateral (Con) skin were analyzed for the expression of indicated markers by immunofluorescent staining. Total number and frequency of cells expressing the indicated markers were determined for each complete biopsy (**A, B** and **E**; data of individual subjects are shown in specific colors) and representative images are shown (**C, D, F** and **G**). Wilcoxon paired tests were used to compare HZ to con samples at the same time point: unadjusted 2-tailed *p* values are shown where *p* ≤ 0.05. **C** Donor 91, day 45 (D45). Scale bar = 50 μm. **D** Donor 78, D90, rash site. Scale bar = 20 μm. **F** Donor 84, D90, rash site. Scale bar = 20 μm. **G** Donor 91, D90, rash site. Scale bar = 20 μm. TIA-1, T cell-Restricted Intracellular Antigen-1; FOXP3, forkhead box P3; T_RM, resident-memory T cells. Number of subjects tested per group: **A** D45 CON, D45 HZ, D90 CON, and D90 HZ, *n* = 7; D360 CON and D360 HZ, *n* = 3. **B** D45 CON, n = 6; D45 HZ, D90 CON, and D90 HZ, *n* = 7; D360 CON and D360 HZ, *n* = 3. **C** D45 CON, *n* = 6; D45 HZ, D90 CON, *n* = 7; and D90 HZ, *n* = 7; D360 CON and D360 HZ, *n* = 3. **D** D45 CON, n = 4; D45 HZ, *n* = 5; D90 CON and D90 HZ, *n* = 7; D360 CON, n = 3; D360 HZ, *n* = 2. **E** D45 CON, *n* = 6; D45 HZ, D90 CON, and D90 HZ, *n* = 7; D360 CON and D360 HZ, *n* = 2. **F** D45 CON, D45 HZ, D90 CON, and D90 HZ, *n* = 7; D360 CON and D360 HZ, *n* = 3. **G.** D45 CON, *n* = 5; D45 HZ, D90 CON, and D90 HZ, *n* = 7; D360 CON and D360 HZ, *n* = 3. Source data are provided as a Source Data file.

screen, Supplementary Fig. 4A). Thirty-one antigenic peptides, representing 26 non-overlapping epitopes for skin CD8 T cells, were identified (Supplementary Fig. 4B; Supplementary Table 5). Eighteen unique skin-CD4 T cell epitopes were also mapped including HLA restricting alleles (representative data, Supplementary Fig. 4C, D). At least one CD4 T cell epitope, ORF9(185–197), was restricted by multiple HLA alleles (Supplementary Table 5).

Next, we estimated the temporal abundance and stability of VZV-specific CD8 T_RM at the epitope and clonotype levels. ICS, used as a semi-quantitative measure of peptide-specificity in skin TCL (Fig. 5A), confirmed CD8 T cell epitope-specificity in HZ-TCL. Amongst CD8 T cells, a median 0.71% of T cells were peptide-reactive, with values generally higher than those in paired CON-TCL (median 0.06%, range 0–13%). The frequencies

declined over time (e.g. HZ day 45, median 4.3% [95% CI 0.8–18.7] compared to HZ day 90, median 0.7% [95% CI 0–3.6]). Among 12 peptides tested on skin TCL from 8 individuals, 11 activated >0.1% of T cells from day 45 HZ-TCL, compared to 5 peptides tested on day 45 CON-TCL. By day 90, 9 peptides remained positive (>0.1%) for the rash site, declining to 4 by day 360. From the rash site, a single day 500 TCL had 3 positive epitope responses. These functional data confirm the long-term presence of VZV-specific CD8 T cell responses at the epitope level in healed HZ skin for many subjects. Curiously, subjects 83, 84 and 90 had at least one contralateral response level that exceeded the rash site, and subject 90 had higher ORF9-specific T cells at day 360 than observed at day 45 (Fig. 5A).

To study the presence and location of these epitope-specific CD8 T cells in skin, we performed in situ tetramer staining on skin biopsies.

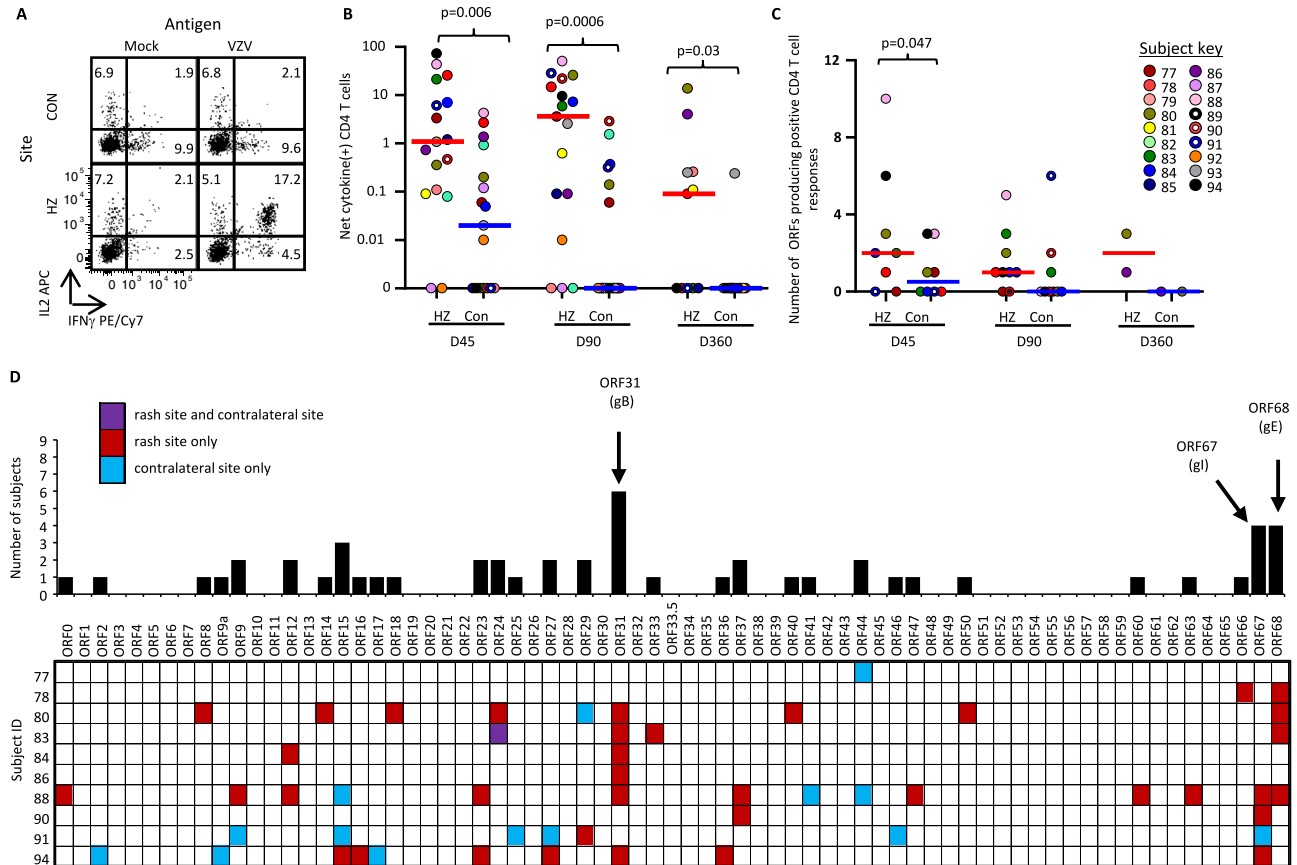

**Fig. 3 | VZV-specific skin-CD4 T cells. A** Example of VZV-specific CD4 T cell detection by ICS for IFN-γ and/or IL-2 in skin T cell lines (TCL) exposed to UV-inactivated VZV (UV-VZV) or mock antigen, showing VZV-specific CD4 T cells localized to HZ skin. Day 90 sample from subject 78. **B** Net frequency of VZV-specific CD4 T cells in HZ-TCL and CON-TCL by ICS for IFN-γ and/or IL-2 (day 45 and day 90 n = 17; day 360, n = 10). Line represents median. Unadjusted 2-sided p values from multiple Wilcoxon tests to compare HZ skin site and CON skin site data within time points are shown. **C** Breadth of T cell proliferative responses as measured by number of reactive VZV ORFs in skin-TCL from days 45 (n = 8), 90 (n = 9), and 360 (n = 2). Skin-TCL pairs were screened if either the HZ-TCL or CON-TCL had >3% net

cytokine-positive cells in ICS assays. Responses in paired cell lines were compared using Wilcoxon's test: Unadjusted 2-sided p values <0.05 are indicated. **D** Summary of the subject-level ORF specificity of proliferative T cell responses in HZ-TCL and CON-TCL from 10 subjects. Each subject studied is shown in a row; each VZV ORF occupies a column. Data are condensed with a response in one or more time point for a subject marked as positive. VZV varicella-zoster virus, ICS intracellular cytokine staining, HZ herpes zoster rash, Con contralateral, D45 day 45, D90 day 90, D360 day 360, ORF open-reading frame. Source data are provided as a Source Data file.

Staining with HLA-A*03:01 tetramers containing the ORF9(111–121) peptide AVYENPLSVEK confirmed that VZV-specific CD8 T cells were present in post-HZ dermal skin at day 45 (donor 91) and day 90 (donor 90) (Fig. 5B), consistent with regions abundant in cells with $T_{RM}$ phenotypes (Fig. 2B, D).

Finally, we determined the clonality and temporal pattern of the epitope-specific CD8 T cells in PBMC and skin by TCRβ sequencing. VZV-specific TCRβ CDR3 clonotypes were determined for CD8 T cells sort-purified from skin-TCL (Subject 79) using four distinct HLA-A*02:01 tetramers. We observed discrete populations of tetramer-specific CD8 T cells in HZ-TCL and, of interest, low level ORF9 tetramer-specific T cells in CON-TCL (Fig. 5C). Temporal changes in total (Supplementary Table 6) and matching epitope-specific TCRβ CDR3 sequences, considered a T cell clone's barcode, were tracked in ex vivo skin biopsy and PBMC samples (Supplementary Table 7, Fig. 5C). As expected, total TCRβ CDR3 reads and diversity were most abundant in HZ skin at day 45 consistent with lymphocytic infiltration. Total TCRβ CDR3 read number and clonotype diversity in CON biopsies were lower than for HZ biopsies at day 45 and day 90, but not at day 360. Multiple distinct VZV-specific TCRβ CDR3 sequences (at the amino acid level) persisted over time in the day 45, day 90 and day 360 HZ sites and were enriched (day 45, 18-fold; day 90, 10-fold; day 360, 7-fold) relative to the CON sites at each time point (Fig. 5D;

Supplementary Table 7). Tetramer-purified CD8 T cell clonotypes matched 27.9% of all T cells detected in the day 45 HZ skin but only 1.6% of T cells in the CON biopsy. VZV-specific T cells declined over time locally at the HZ site (20.8% at day 90, 9.1% at day 360), but remained elevated relative to non-involved skin (2.1% at day 90; 1.3% at day 360). VZV-specific CD8 T cells were also enriched at the rash site compared to blood at all time points.

Although diverse—with 23 and 68 clonotypes detected, respectively (Supplementary Table 7)—the top 5 clonotypes for ORF9 and ORF18 occupied >90% of the within epitope-specific responses at day 45 at the rash site, reflecting clonal immunodominance. Fewer clonotypes were identified for ORF53 (n = 4) and ORF62 (n = 2). The dominant CDR3 sequences for ORF9 and ORF18 gradually declined in frequency over time yet remained higher in HZ skin relative to CON skin and PBMC, even after one year (Fig. 5E, Supplementary Table 6). Each ORF53 clonotype was enriched in HZ skin relative to PBMC at each time point. An ORF53-specific CD8 T cell clonotype that emerged as dominant at day 360 was subdominant at day 45 (Fig. 5E), consistent with functional proteome screening assays at this time point (Supplementary Fig. 2B). Despite recovery of ORF62 T cells from day 360 HZ skin (Supplementary Figs. 2B and 4B, Fig. 5), levels of corresponding TCRβ sequences in the separate HZ biopsy and PBMC samples were low, suggesting ORF62-specific CD8 T cells are rarer than

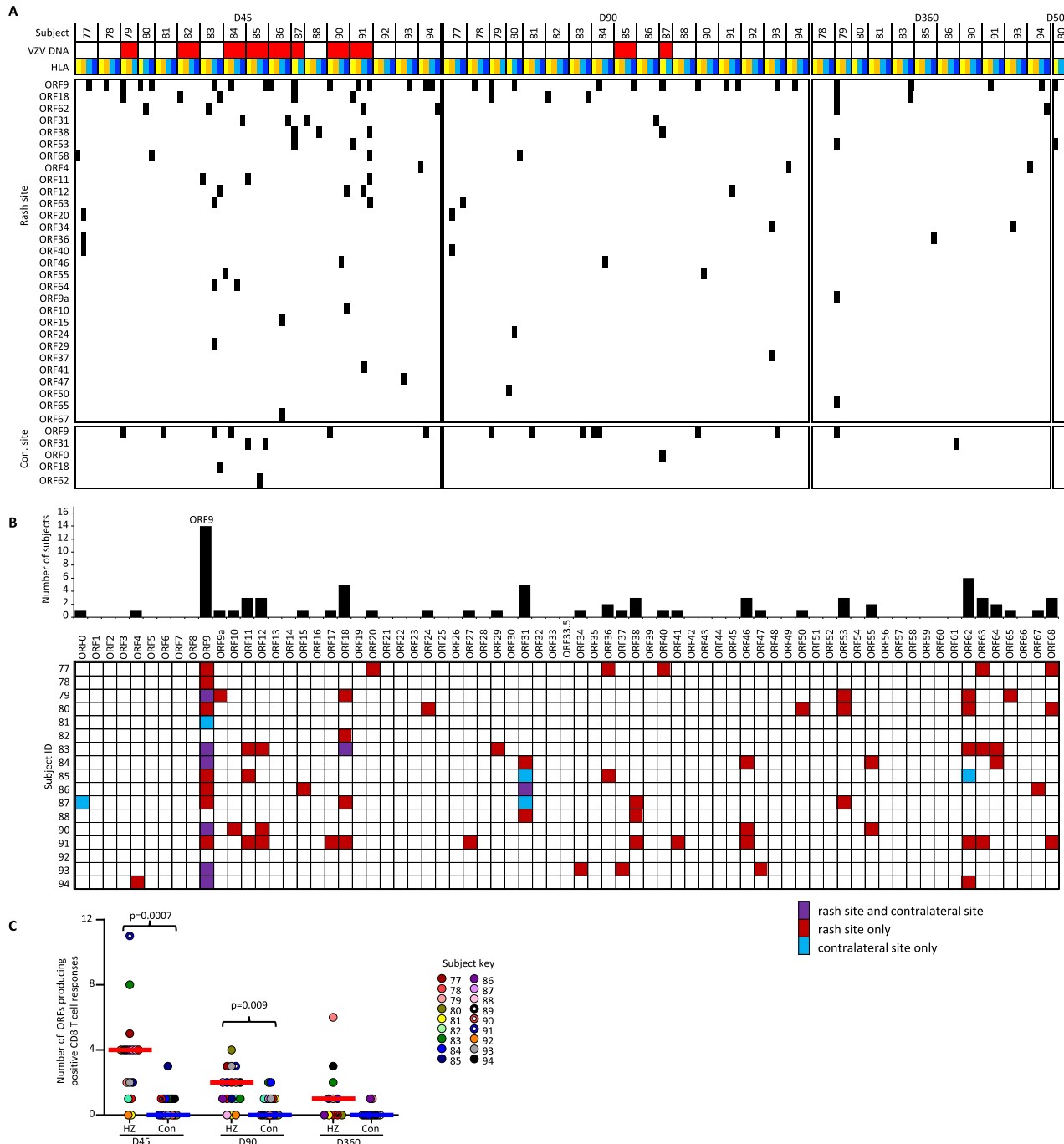

**Fig. 4 | VZV-specific skin CD8 T cells.** Skin-TCL containing a mixture of CD4 and CD8 T cells were generated from post-HZ and unaffected contralateral skin site as outlined in the methods. **A.** Heatmap showing positive CD8 T cell responses (black boxes) from each rash or contralateral biopsy to artificial antigen presenting cells co-expressing an individual subject-specific HLA class I (columns) and single VZV ORF (rows). Days after HZ rash onset are shown at the top with subject numbers underneath. Detection of VZV DNA at the time of biopsy is indicated with red. HLA-allele color codes: HLA-A in light or dark yellow, HLA-B in light or dark blue. Only VZV ORFs with at least one positive response are shown. **B** Summary of the per subject specificity of CD8 T cell responses to each VZV ORF in biopsies obtained from the rash and/or contralateral skin from 17 subjects. Data are condensed, such that the responses were observed at any of the tested time points and across all subject-specific HLAs. Biopsy locations are color-coded for HZ, contralateral, or both biopsies. **C** The number of VZV ORFs that trigger CD8 T cell responses within each skin-TCL integrated across all HLAs within each subject. Each dot represents a single biopsy. D45, *n* = 17 D90, *n* = 17, D360, *n* = 11. Responses in paired skin-TCL were compared using Wilcoxon's test: Unadjusted 2-sided *p* values <0.05 are indicated. VZV varicella-zoster virus, HZ herpes zoster rash, Con contralateral, D45 day 45, D90 day 90, D360 day 360, D500 day 500, ORF open-reading frame, HLA human leukocyte antigen, TCL T cell line. Source data are provided as a Source Data file.

those of other ORFs. The dominant clonotypes were always enriched in HZ site skin compared to CON skin or PBMC, reinforcing these VZV-specific CD8 T cell clonotypes as T$_{RM}$ that preferentially persist in HZ skin.

## Discussion

T cells are a functionally important arm of the acquired immune response during HZ[28]. We measured VZV-specific CD8 and CD4 T cells in rash site skin biopsies in otherwise healthy persons, with

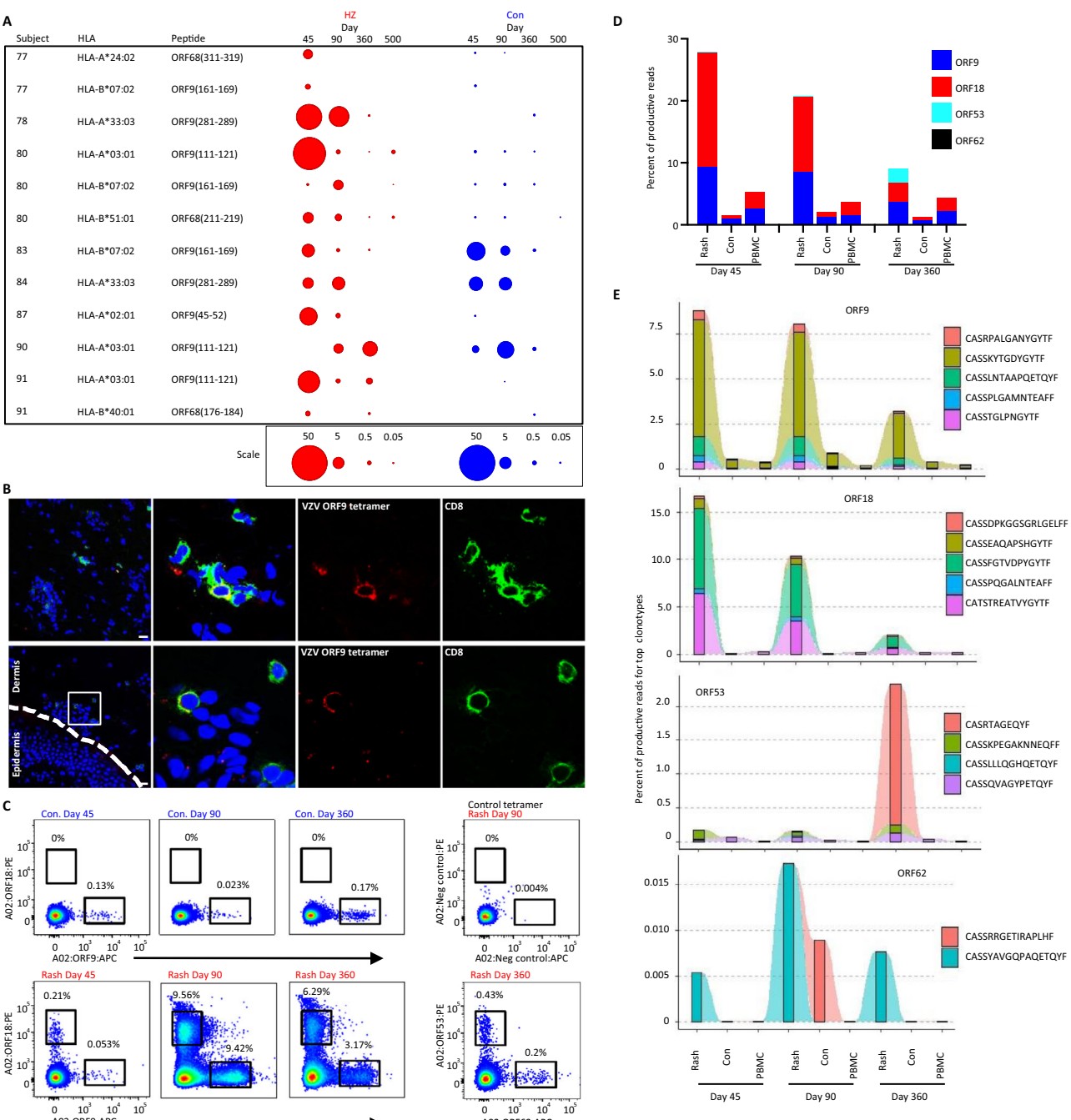

**Fig. 5 | VZV-specific CD8 T cell clonotypes persist long-term in post-zoster skin.** **A** Bubble plots of the percentage of CD8 T cells in skin-TCL reactive to the indicated peptide epitopes show VZV-specific CD8 T cell persistence well after herpes zoster resolution. Frequencies represent net IFNγ+ events amongst all live CD3+ lymphocytes calculated by subtracting IFNγ+ events in DMSO-exposed controls. Bubbles sizes are scaled as indicated at bottom. **B** In situ staining of post-HZ skin biopsies confirm the presence of HLA-A*03:01-ORF9-tetramer (peptide AVYENPLSVEK) positive CD8 T cells in the dermis. Top images are for subject 91, day 45 (VZV DNA positive) HZ biopsy; bottom images are for subject 90, day 90 (VZV DNA negative) HZ biopsy. **C** HLA-A*02:01 tetramers specific for distinct VZV epitopes were used to isolate VZV-specific CD8 T cells from the T cell lines of subject 79 for *TRB* sequencing after DNA isolation. **D** Skin biopsy and PBMC

samples for each time point from subject 79 were used for *TRB* sequencing. The frequency of productive *TRB* CDR3 sequences in ex vivo samples that matched to tetramer-sorted T cells are shown for each epitope. Stacked bars show the total collective response for the 4 epitopes over time within sample. Each set of TRB sequences from tetramer-purified CD8 T cells is represented by a different color according to the key. **E** Tracking of the top 5 ranked clonotypes based on the frequency in day 45 rash biopsy samples confirm one or two clonotypes dominate the response for each ORF and remain higher in HZ versus contralateral biopsies in later time points. VZV varicella-zoster virus, HZ herpes zoster rash site, Con contralateral skin site, ORF open-reading frame, HLA human leukocyte antigen, TCL T cell line, TRB T cell receptor β. Source data are provided as a Source Data file.

contralateral skin providing within-subject controls. Rather than broad dissemination throughout the skin, we found that VZV-specific T cells were selectively localized to skin at rash sites at day 45 after rash onset and, notably, at later time points when virus was no longer detectable

and systemic immune responses had considerably declined. In this report, we demonstrate that VZV-specific T cells persist as $T_{RM}$, and thus are seeded and retained in a spatially-restricted manner by the inflammatory and repair events that take place during and after HZ.

Our year-long study of post-HZ skin offers novel insight into the $T_{RM}$ life cycle in human skin. It is likely that skin seeding of early, infiltrating effector cells occurs during childhood varicella. However, detailed information concerning skin $T_{RM}$ in the decades between varicella and HZ is limited. Our current study now documents that local persistence of long-term virus-specific $T_{RM}$ favors sites of antigen re-challenge during HZ. T cells migrating to HZ skin displayed CD69 and CD103, considered as canonical $T_{RM}$ markers[1,29]. These cells appeared more cytolytic during early time points than later time points post-HZ, reminiscent of an effector response transitioning to a quiescent, poised state. In comparison, there was only limited infiltration noted to paired control skin sites. Brisk localization of T cells to HZ skin at day 45 was not unexpected. In the context of HSV infection, tissue-specific migration and spatial enrichment of CD4 and CD8 T cells with antiviral and cytotoxic activity is well-documented for skin and eye lesions[30,31]. However, repetitive HSV shedding at the same anatomic locations[32] make it difficult to discern long-term $T_{RM}$ from recirculating T cells or inflammatory infiltrates at HSV lesion sites. HZ recurrences and shedding, in contrast, are uncommon[19] implying VZV antigens and associated inflammation are not intermittently or continuously present in skin. Thus, consistent with bona fide $T_{RM}$, long-term T cell persistence after HZ is independent from intermittent or sustained antigen presence. The higher T cell abundance we observed in the dermis than in the epidermis of HZ skin corresponds positionally to VZV protein expression during HZ[33]. In contrast, studies of HSV infection show CD8 T cells with antiviral and $T_{RM}$-associated biomarkers in the epidermal layer affected by HSV infection, with CD4 $T_{RM}$ occurring deeper[16,34]. These observations imply that the programming of virus-specific skin-infiltrating CD8 T cells to remain as $T_{RM}$, and their fine anatomic positioning in the particular region(s) of viral reactivation in skin, are both important. We hypothesize that the precise placement of VZV-specific $T_{RM}$ may explain why HZ seldom recurs in the same dermatome.

We demonstrated that $T_{RM}$ persisting in HZ skin were VZV-specific. Our techniques largely prevented ex vivo quantitation of VZV-specific skin-CD4 T cells. However, we found a major contribution of VZV-specific CD4 T cells in total skin-resident cells, at least in some subjects, consistent with studies employing an intradermal challenge to recall VZV-specific CD4 T cells[35]. We recognize that some infiltrating CD4 T cells may not be VZV-specific and serve other roles, such as extinguishing inflammation or wound healing[36], consistent with the $T_{reg}$-like signature we observed in later biopsies and aligning with the prior short-term observations during VZV antigen challenge[35]. CD8 and CD4 T cells with VZV-specificity were both selectively enriched in cultures expanded from the early HZ site relative to control skin. This agrees with a previous investigation wherein VZV-specific T cells from both subsets were boosted in skin injected with whole VZV antigen but not in unchallenged skin[37]. For a few defined epitopes, the in situ frequency of VZV-specific CD8 $T_{RM}$ was estimated at about 10% of rash site T cells one year after HZ, with marked clonotypic dominance. These levels of VZV-specific CD8 $T_{RM}$ in HZ skin much exceeded those in PBMC and uninvolved skin, highlighting a very localized retention of VZV-specific $T_{RM}$ following HZ rather than the wide dissemination expected during varicella. Extended enrichment of VZV-specific T cells at the rash site is consistent with reduced efflux of $T_{RM}$ once established in the skin, particularly for CD8 $T_{RM}$[29].

The functional mechanisms of VZV-specific T cells at infected sites remain to be fully elucidated. Their potential to clear virus or promote tissue healing was not investigated in this report because we started biopsies after lesion resolution. However, based on data from active human HSV lesions[30,38], it is reasonable to propose that recruitment and/or in situ proliferation of VZV-specific CD4 and CD8 T cells occurs in skin during active HZ. Future research using HZ vesicle fluid or biopsy of active lesions combined with knowledge of VZV-reactive T cells determined via TCR sequencing or TCR databases[39,40], could

facilitate further interrogation of the contribution made by infiltrating, resident, and recirculating[41] VZV-specific T cells to viral clearance, wound healing, and resolution of HZ lesions. The pro-inflammatory effects of VZV-specific T cells may contribute to lesion and HZ pain, severity and duration. Human autopsy studies reveal the trigeminal ganglia, a site of VZV latency and reactivation, contains T cells abutting neurons after the resolution of HZ supporting a potential for local antigen-reactive T cells to contribute to PHN pathogenesis[42]. VZV-specific T cells persisting after HZ resolution may also aid host defense against future episodes of HZ in the same dermatome, as seen in murine studies with HSV, whereby local skin CD8 $T_{RM}$ provide anatomically restricted protection against reinfection[43]. Unfortunately, current animal models of VZV infection[44] are unlikely to contribute to functional studies of local VZV-specific $T_{RM}$. Natural history investigations of local protection in human subjects against second episodes of HZ are also very challenging given the overall low incidence of recurrent HZ[19]. As a surrogate for HZ, the recently described model system in which inactivated VZV antigen is introduced into human skin[35] could be used to assess differences between healed HZ and control sites, with a focus on CD4 T cells. Use of replication-competent, attenuated VZV strain vOKA, licensed as a vaccine, might be a better future experimental medicine approach to study CD8 $T_{RM}$ in human skin. It is not known if the highly clinically effective recombinant zoster vaccine increases skin-resident VZV-specific T cells. Such a finding would be consistent with, but not prove, a role for $T_{RM}$ in the mechanism of action of this vaccine.

One interesting observation was the recovery of VZV-specific CD8 T cells from uninvolved skin, albeit at lower prevalence, frequency, and antigenic diversity, than for rash sites. There are several possible explanations. The most plausible is that VZV-specific T cells widely disseminated to skin during varicella, decades earlier, remain detectable in non-HZ skin. Another study described VZV-specific CD4 T cells in the skin of aged subjects with a history of varicella but not HZ[45], suggesting VZV-specific T cells are long-lived in skin. However, the presence of VZV-specific CD8 T cells in uninvolved skin has yet to be verified in the absence of recent HZ. An alternate hypothesis is that boosted systemic VZV-specific CD8 T cells distribute broadly throughout the skin due to transient expression of the skin-specific adhesion molecule cutaneous leukocyte antigen (CLA)[46]. It is also possible that VZV-specific CD8 T cells exit the rash site, circulate, and lodge elsewhere, consistent with the recent description of circulating $T_{RM}$-like cells[29,41]. Similar mechanisms are supported by evidence that blood HSV-1 and HSV-2-specific memory CD8 T cells are permanently programmed to express high levels of CLA and chemokine receptors relevant to skin homing[47].

Our proteome-wide studies revealed that skin-CD4 T cells frequently recognize VZV glycoproteins, while both skin and blood CD8 T cells focus on ORF9[26,28]. Other studies have shown skin T cells have diverse TCRs[16,48], but rarely document antigen specificity of $T_{RM}$. We observed some temporally persisting T cell clonotypes were consistent in assigned specificity to VZV epitopes and HLA restriction in our study, further supporting our hypothesis that virus-specific T cells recruited to clear VZV are programmable for long-term retention. The high population prevalence of ORF9 make it an intriguing antigen candidate for vaccine approaches aimed at recruiting antiviral CD8 T cells to the skin and retaining them as protective $T_{RM}$.

There are some limitations to our study. First, it has been reported that some $T_{RM}$ subsets show reduced proliferation in culture. We expanded diverse, robust, virus-specific T cells, suggesting that some VZV-specific $T_{RM}$ have renewal capacity[49], but may have missed antigen-specific $T_{RM}$ with poor proliferation. Second, due to clinical concerns, our studies were temporally limited and missed the anticipated peak magnitude and diversity of an early T cell response. Additionally, under-sampling, inherent in small biopsies, potentially underestimated presence or diversity of $T_{RM}$[50]. Heterogeneity with

regards to HSV infection may alter systemic T cell responses to VZV due to cross-reactivity[13,14,39]. Some subjects declined the final biopsy, leading to a lower number of observations at the later time points that might make our conclusions less certain by day 360. HZ is classically a disease of the aged and many of our subjects were relatively young. The median subject age of 47 is close the US-recommended age (50) for receiving the recombinant zoster vaccine. This cutoff was set based on the age of an upswing in HZ risk[51]. We observed no correlation between age and HZ-site skin-CD4 or CD8 T cell responses within our small cohort. Future studies could uncover potential differences in VZV-specific $T_{RM}$ with aging. Our case-finding method was unbiased regarding age. During recruitment for this study (2018–2020), uptake for shingles preventative vaccination was low in the US. Therefore, it is unlikely that our study was biased to younger persons related to low HZ rates in older persons. We note a few participants in our study received shingles vaccination, but we are unable to assess whether this impacted the results. It is also possible that some CDR3 sequences may represent contaminants with specificity to other pathogens. Amongst VZV ORF18 tetramer-positive cells, we noted one low frequency sequence (CASSLAPGATNEKLFF) associated with CMV-associated memory inflation[52] as a possible contaminating TCR-β, but no other such sequences were detected.

In conclusion, we investigated the localization of pathogen-specific T cells in human skin. We observed that in response to a single, intense burst of viral protein expression during VZV reactivation, virus-specific CD4 and CD8 T cells both robustly and selectively localized to sites of HZ rash in immunocompetent humans for at least one year. Within-subject temporal consistency for fine specificity and TCR CDR3 sequences, and tetramer in situ straining, support long-term retention of infiltrating T cells as $T_{RM}$. HZ provides a unique, accessible model system with a mature set of T cell tracking tools to decipher the natural history of antigen-specific $T_{RM}$ in humans.

## Methods

### Subjects and specimens

Subjects with HZ were recruited to University of Washington Virology Research Clinic approximately one month after HZ onset using the electronic health record query AMALGA system to identify participants[53]. The day of self-reported rash onset was defined as day 0. Overt rash was absent by day 45. At the initial visit (day 30), a swab was performed at the site of HZ for VZV DNA. Blood and two 3 mm diameter skin biopsies from both the rash and contralateral sites were obtained at day 45 and day 90 (range 77–111) post-HZ onset, with an optional biopsy at day 360 (range 351–388) and day 500. The day 90 biopsy window was 77–111 days after HZ onset, and the one-year windows was 351–388 days. Biopsies were performed at subject-described areas of prior HZ or the contralateral side within the same dermatome. Pre- and post-biopsy photographs and areas of post-biopsy hypo- or hyper-pigmentation, if present, were used to guide the locations of sequential biopsies as closely together as possible. Swabs were obtained prior to biopsy for quantitative VZV[54] and HSV PCR[55]. Peripheral blood mononuclear cells (PBMC) were isolated within 8 h of venipuncture using Lymphoprep (Cosmo Bio USA) and cryopreserved at $10 × 10^6$ cells/mL/vial in 10% DMSO (Fisher), 40% human serum (Valley Biomedical), and 50% RPMI-1640 (Hyclone). Serum was stored at −80 °C. The study was approved by the University of Washington Institutional Review Board. Participants provided written informed consent. For studying VZV-specific T cells in blood, a separate cohort of healthy donors was recruited at the Sanquin-National Blood Bank of The Netherlands (Amsterdam). After informed consent, leukapheresis was performed and subjects seropositive for anti-VZV IgG were HLA typed. PBMC were similarly isolated and cryopreserved in aliquots.

B lymphocyte continuous lines (B-LCL) were immortalized from PBMC using EBV strain B95-8[56]. One skin biopsy from each site per subject was rinsed in PBS, snap frozen in a 2 mL tube by submersion in liquid nitrogen and stored at −80 °C. The other biopsy was used to generate bulk skin T cell lines containing both CD4 and CD8 T cells. To stimulate T cell outgrowth from skin, biopsies were scalpel-minced in 1 mL T cell medium (TCM) containing RPMI-1640, 4% human serum, 4% defined fetal bovine serum (Hyclone), 2 mM L-glutamine (Hyclone), and 100 U/mL penicillin/streptomycin (Gibco) and cultured in two wells of a 48-well plate each containing $2 × 10^6$/mL γ-irradiated (3300 rads) allogeneic PBMC and 1.6 μg/mL PHA-P (Remel) for 2–3 weeks (37 °C, 5% $CO_2$). Natural IL-2 (nIL-2, 32 U/mL, Hemagen) was added after 48 h and replenished with fresh TCM twice weekly. Polyclonal T cell lines were further expanded using anti-CD3[57] and resultant T cell lines cryopreserved in aliquots.

### Antigens

For whole viral antigen, VZV (strain vOka, Zostavax™, Merck) was propagated on human embryonic tonsil (HET) cells in Eagle's modified essential medium (EMEM; Corning) containing 10% fetal bovine serum (FBS), 2 mM L-glutamine, and 1% penicillin/streptomycin (all Hyclone) as described[26]. At >50% cytopathic effect, cells were removed by scraping, resuspended in complete EMEM, sonicated, and centrifuged ($400 × g$, 10 min). Supernatant (100 μL droplets) was exposed to UV light for 30 min, pooled, aliquoted, and frozen as ultraviolet (UV)−inactivated VZV antigen (UV-VZV). Mock UV-HET antigen was made in parallel.

To generate cell-free VZV (strain Webster) stocks, confluent primary human lung fibroblasts (HLF, ATCC, Cat # CCL-199) were co-cultured 5:1 with VZV-infected cells in MEM supplemented with 10% FBS at 37 °C, 5% $CO_2$. Infected cells were harvested when CPE reached 80% (about 2 days), disrupted by sonication, debris removed by centrifugation at 4 °C, and the harvested supernatant aliquoted and stored in −70 °C.

Each VZV open-reading frame (ORF) was cloned into Gateway™ pDONR207 or pDONR221 (Invitrogen, Grand Island, New York) as described[26]. ORFs were subcloned into pDEST203[58] for CD4 T cell assays. pDEST203-VZV plasmids were used to express VZV proteins as N-terminal 6-His fusions via *E. coli* in vitro transcription translation (IVTT) (Expressway, Invitrogen). Negative-control IVTT products were made from *M. tuberculosis* genes[25] and empty pDEST203 vector. For CD8 T cell assays, VZV ORFs were subcloned to pDEST103[58] for eukaryotic intracellular expression of VZV-eGFP fusion proteins.

Peptides (Genscript, >70% pure) were reconstituted in DMSO and used either pooled or separately at a final concentration 1 μg/mL each. DMSO in assays was <0.3%. Peptide sets contained 20-mers overlapping by 11-amino acids or 13-mers overlapping by 9-amino acids spanning VZV ORFs of the Dumas or vOka sequence. Shorter peptides (8–10 amino acids) were designed using algorithmic prediction of HLA binding[59] within antigenic 13-mers.

Since VZV shows limited sequence variation[60], we anticipated the viral antigens used for all aspects of this study should be adequate to recall T cell responses in persons whose viral sequences were unknown.

### Assay of VZV-specific immunity in blood

VZV-specific serum IgG antibodies were measured by ELISA using mixed VZV glycoprotein (VZVgp) antigen[61]. In brief, plates were coated with VZVgp and uninfected tissue control antigens (Merck and Co. via material transfer agreement) at 1 μg/mL in PBS for 18–24 h at 4 °C. After washes and blocking with 5% skim milk for 30 min at room temperature, serial (1:20) dilutions of sera were added in duplicate at 100 μL/well for 30 min at room temperature. After aspiration and washes with PBS/Tween-20, goat anti-human IgG-alkaline phosphatase conjugate (SeraCare, 5220-0351 (0751-1006)) diluted 1:1000 in PBS was added for 30 min. Disodium nitrophenyl phosphate (Sigma) substrate was used per the manufacturer with addition of 100 μL 2.5 N NaOH stop solution at 10 min. Plates were read at 405 nM and adjusted OD reported as mean test OD minus mean control OD.

For virus neutralization, 200 μL HLF cells ($1.25 \times 10^5$ cells/mL in MEM-10% FBS) were added to each well of an Immunlon 96-well sterile culture plate (Dynex Technologies, Inc.) and incubated overnight in a humidified incubator (37 °C, 5% CO₂). All test sera were heat-inactivated 30 min at 56 °C to inactivate complement. Cell-free VZV (25 pfu) was added to a 2-fold dilution series of test sera in MEM-10% FCS and incubated (37 °C, 5% CO₂) for 30 min. Guinea-pig complement (Sigma; S1639) was added at 2.5 U/well and the serum-VZV-complement mixture incubated for a further 30 min at 37 °C, 5% CO2. The serum-VZV-complement series was transferred to the appropriate wells of the pre-prepared HLF plate after removal of media. Positive and negative-control sera were included as assay controls. After incubating (37 °C, 5% CO₂) for 60 min, the serum-VZV-complement cocktail was replaced with 150 μL MEM-10% FCS per well and the plate incubated for 3 further days at 37 °C. The HLF monolayer was washed with 1× PBS (pH7.2, Gibco), fixed in cold methanol, and washed once more with PBS. Non-specific binding sites were blocked at room temperature for 30 min using BLOTTO (PBS/0.1% Tween with 5% skim milk) and VZV detected by incubation with anti-VZV gE Mab (Clone M1, Millipore, #MAB8612) diluted 1:1000 in BLOTTO followed by goat anti-mouse IgG/ HRP (Invitrogen, G21040) conjugate diluted 1:1000 in BLOTTO (30 min each at room temperature). Bound antibody was detected using TMB substrate (development stopped using 1.0 N H₂SO₄) with optical density (OD) measured at 450 nm.

VZV-specific CD4 T cells were quantified in PBMC by intracellular cytokine staining (ICS) flow cytometry[26]. After overnight rest in TCM, $10^6$ PBMC were incubated for 18 h at 37 °C with costimulatory antibodies (anti-CD28 and anti-CD49d; BD Biosciences) and UV-VZV or UV-mock (1:100), medium, or PHA-P (1.6 μg/mL) in TCM: Brefeldin-A (Sigma) was added after 2 h. After Live/Dead Fixable Near-IR (Invitrogen) stain, cells were treated with FACS Lysing and Permeabilizing 2 solutions (BD) and stained with anti-CD3-ECD (UCHT1, Beckman Coulter), anti-CD4–PE (A161A1; Biolegend) or anti-CD4-FITC (S3.5; Invitrogen), anti-CD8-FITC (3B5, Invitrogen) or anti-CD8-PerCP/Cy5.5 (SK1; BD), anti-IFN-γ-PE/Cy7 (4S.B3; BD), and anti-IL-2-APC (MQ1-17H12; BD). Data collected on a BD LSRII cytometer using BD FacsDiva Version 8.0.1, and were analyzed with FlowJo v10 (BD). Single live CD3+CD4+CD8− lymphocytes were assessed for IFN-γ and IL-2. The gating strategy is shown in Supplementary Fig. 5A.

In the Netherlands blood bank cohort, polyclonal VZV-specific CD8 T cells were enriched by CD137-based activation induced marker (AIM) sorting modeled after HSV-1 work[58]. In brief, MRC-5 cells (ATCC CCL-171), infected with VZV strain Dumas at MOI 0.01 for 3 days, were scraped into a small volume of PBS, treated with UV-C light (0.12 mJ/cm², Vilber Lourmat BioLink BLX-254 Crosslinker) and added to monocyte-origin dendritic cells (moDC) prepared from PBMC[58]. Autologous CD8 T cells enriched from PBMC by negative immunomagnetic depletion (Miltenyi) were added; conditions were as published[58]. Cells were stained after 18 h with αCD3-APC/Cy7 (SP34-2, BD), αCD8 PE/Cy7 (RPA-T8 eBioscience), αCD4-PerCP (SK3, BD), and αCD137-PE (4B4-1, Miltenyi) and CD3+CD8+CD137+ cells were sorted and polyclonally expanded. Resulting CD8 T cell effector lines were screened for reactivity to artificial antigen presenting cells (aAPC) made by co-transfection of COS-7 cells with plasmids encoding subject-specific HLA class I cDNA and VZV ORFs as published and outlined below[38,58].

### In situ analyses of skin T cells

Immunofluorescent staining was performed on acetone-fixed 8 μm-thick cryosections of the snap frozen skin biopsies, as described[62]. In brief, sections were blocked and incubated overnight at 4 °C with the following primary antibodies, diluted in PBS containing 1.0% bovine serum albumin (BSA): mouse anti-CD4 (clone 4B12; dilution 1:40; Dako), anti-CD8 (1A5; dilution 1:40; Monosan), anti-CD69 (FN50; dilution 1:100; Biolegend), anti-CD103 (2G5-1; dilution 1:25;

Thermo Fisher), anti-Ki67 (MIB-1; dilution 1:100; Dako), anti-PD-1 (EH33; dilution 1:150; Cell Signaling Technology), anti-TIA-1 (2G9A10F5; dilution 1:100; Beckman Coulter), anti-VZV IE63 (9D12; dilution 1:500; gift from Catherine Sadzot-Delvaux, University of Liege, Belgium), rabbit anti-CD3ε (SP7; dilution 1:100; Thermo Fisher), rat anti-CD8αβ (YTC182.20; dilution 1:200; Bio-Rad) and anti-FoxP3 (PCH101; dilution 1:20; Thermo Fisher). Sections were stained with the following secondary antibodies diluted in PBS containing 0.1% BSA for 1 h at room temperature (RT): Alexa Fluor 488 (AF488)-conjugated goat anti-mouse IgG1 and anti-rat IgG, AF594-conjugated goat anti-mouse IgG2a and anti-rabbit IgG, AF647-conjugated chicken anti-rabbit IgG and goat anti-mouse IgG1 (all dilution 1:250; Thermo Fisher). Nuclei were stained with Hoechst-33342 (Sigma Aldrich) and sections were mounted in Pro-long diamond antifade reagent (Thermo Fisher). Slides were scanned using a Zeiss Axio Imager II and images were analyzed using QuPath version 0.2.3[63].

In situ tetramer staining was performed using VZV ORF9-specific APC-conjugated HLA-A*03:01 AVYENPLSVEK and HLA-A*33:01 QTTGRITNR tetramers (Immunaware), as described[64]. Cryosections (8 μm-thick) were fixed in 4% paraformaldehyde solution in PBS (4% PFA), washed twice with PBS, blocked for 30 min at RT using 5% goat serum diluted in PBS, and incubated overnight at 4 °C with MHC class I tetramers 1:10 diluted in PBS containing 0.1% BSA. Sections were washed twice with PBS and fixed with 4% PFA for 15 min at RT. CD8 staining was performed using mouse anti-CD8 (clone 3B5; dilution 1:50; Thermo Fisher) and AF488-conjugated goat anti-mouse IgG (both Thermo Fisher). Nuclei were stained with Hoechst-33342 (Sigma Aldrich) and sections−mounted in Prolong diamond antifade reagent−analyzed on a Zeiss LSM 700 confocal laser scanning microscope using ZEN 2010 software (Version 6.0, Zeiss) to adjust brightness and contrast.

### VZV-specific CD4 T cell responses in skin

Skin-CD4 T cells reactive to whole VZV was initially assessed using ICS. Biopsy-derived, PHA-expanded T cells ($2.5 \times 10^5$) were combined with equal numbers of cell tracker violet (CTV, Invitrogen)−stained autologous PBMC and antigens described for blood assays, above. Staining conditions were already described. Using FlowJo, CTV + PBMC were dump-gated and the frequencies of live CD3+CD4+CD8− T cells expressing IFN-γ and/or IL-2 were determined. The gating strategy is shown in Supplementary Fig. 5B. When HZ biopsies had >3% cytokine-positive CD4 T cells (net) after VZV exposure, responses to individual VZV proteins were determined for both contralateral and HZ skin-derived T cell lines as follows. T cells and γ-irradiated (3300 rad) autologous PBMC ($10^5$ each) were combined in 200 μL TCM, in duplicate round-bottom wells, with VZV or control proteins at 1:2000 final dilution, UV-VZV and UV-mock antigens (1:200), or PHA (1.6 μg/mL). Following 72 h of incubation at 37 °C, cultures were pulsed with ³H-thymidine (0.5 μCi; Perkin Elmer). The next day, cells were harvested onto Unifilter-96 GF/C plates (Perkin Elmer) and ³H-thymidine incorporation determined using Microscint-20 (Perkin Elmer) and Topcount NXT (Perkin Elmer).

To determine antigenic peptides, autologous PBMC or B-LCL ($2.5–5.0 \times 10^4$) were combined with biopsy-derived T cells ($10^5$) and peptide pools (1 μg/mL each peptide)[65]. To determine the restricting HLA locus of reactive peptides, mAb supernatants from mouse hybridomas L243 (anti-DR), SPVL-3 (anti-DQ), B7/21 (anti-DP), or medium alone (negative control) were combined (1:4) with APC prior to addition of T cells and 0.01 μg/mL, 0.1 μg/mL, and 1 μg/mL peptide[66]. To determine restricting HLA alleles, HLA class II single antigen lines (SAL)[67] were used as APC with 1 μg/mL peptide or were pulsed with 10 μg/mL peptide, washed with PBS, and then exposed to T cell lines. Parental DAP.3 or RM3 cells were included as HLA-negative controls. For each assay, supernatants were harvested for IFN-γ ELISA[38] after 24 h incubation of cultures at 37 °C (5% CO₂). Assays were

performed in duplicate or triplicate and data collected on a Victor2 microplate reader using Wallac 1420 Workstation software (version 3, PerkinElmer).

### VZV-specific CD8 T cell responses in skin

Artificial APC (aAPC) expressing subject-specific HLA class I heavy chains and individual VZV ORFs cloned into pDEST103[58] were used to detect CD8 T cell responses in biopsy-derived T cell lines. pDEST103:VZV-ORF plasmids (100 ng/well) and empty p103 negative-control plasmids were co-transfected with pcDNA3.1 constructs encoding subject-specific class I HLA-A or B (100 ng/well) into COS-7 cells in duplicate as described[58]:no-HLA controls were also assessed. T cells ($10^5$) were added 48–72 h post-transfection in TCM and incubated for 24 h at 37 °C. PHA (1.6 µg/mL) was used as a positive control. Secreted IFN-γ was measured in supernatants by ELISA[38].

Epitopes were mapped using methods similar to a previous report[68]. Briefly, COS-7 were transfected with HLA cDNA (100 ng/well). After 48–72 h, cells were either: (1) incubated with $10^5$ biopsy-derived T cells and 1 µg/mL peptide or (2) pulsed with 10 µg/mL peptide, washed twice with PBS, and overlaid with $10^5$ CD8 T cells. Peptides were used as complex pools or as selected HLA-binding motif matched peptides. DMSO was negative control for peptides while full-length transfected VZV ORF, or PHA alone (1.6 µg/mL) were used as positive controls. For selected epitopes, PE- or allophycocyanin-conjugated HLA class I tetramers were constructed (Flex-T, Biolegend, per manufacturer). Tetramers were used to stain T cells ($2 \times 10^6$) for 30 min on ice in the dark, followed by staining with αCD3-ECD (UCHT1, Beckman Coulter), αCD4-BV421 (RPA-T4, BD), αCD8-FITC (3B5; Thermofisher), and 7-AAD (BD). Tetramer-positive cells were sorted (FACS Aria II) into TCM, pelleted, and frozen at −80 °C. The gating strategy is shown in Supplementary Fig. 5C.

### TCR analyses

*TRB* CD3 sequences of tetramer-sorted T cells, whole PBMC, and 3 mm biopsies were performed by Adaptive Biotechnologies (v4b Immunoseq assay). DNA was manually extracted (DNeasy Blood/Tissue Kit, Qiagen) from tetramer-sorted cells or robotically (by Adaptive) for other samples. Data exported from the Adaptive Immunoseq platform were analyzed using VDJTools (v1.2.1)[69] as follows. Data were converted to VDJtools format using the Convert command and pre-processed using Correct to remove erroneous sequences and merge those with ≤2 nucleotide mismatches. Out-of-frame reads were removed using FilterNonFunctional, retaining original clonotypes frequencies. Clonotype tables for all bulk skin and PBMC samples were generated by the JoinSamples function using default parameters and amino acid sequences for matching. For the tetramer-purified samples, within-tetramer CDR3 data for all time points were combined and further filtered to remove CDR3 sequences: (1) that were represented <4 times within combined data and (2) that were present in multiple tetramer-purified fractions unless the CDR3 sequence occupied >10% of only one sample. PBMC and skin biopsy-derived data were filtered to CDR3 sequences identified for each of the four tetramer-sorted skin-derived T cell samples using Microsoft Excel 2010. The top 5 clonotypes were tracked and graphed using the trackClonotypes function within ImmunArch (v0.6.6[70]).

### Statistics

The nonparametric Wilcoxon's test for paired samples was used to compare adjacent time points for VZV DNA, antibody titers, and ex vivo ICS. The Wilcoxon's test was also used to compare T cell breadth or phenotype metrics between HZ and CON skin samples. Pairwise testing was performed due to absence of some subjects in later time points that prohibited equivalent repeated measures testing. Correction for multiple comparisons was not done. Two-sided p values <0.05 were considered statistically significant. Statistical tests and graphs were generated using Prism (v8, GraphPad). Outlier analysis was used for CD4 T cell VZV proteome-wide screens: response to VZV proteins surpassing a threshold (median response of all negative-control antigen samples + 2.33 × median absolute deviation of all negative-control antigens) in lymphoproliferation assays was individually determined per cell line. This method has a theoretical false-positive rate of 1.0%[26]. Antigens were scored positive if both replicates exceeded the threshold. ORFs expressed as fragments (e.g. ORF22) were scored positive once if one or more fragment was positive. For CD8 T cell VZV proteome-wide screens, responses to VZV proteins were deemed positive if both replicates exceeded IFN-γ ELISA $OD_{450}$ values of 0.15 and were at least twice the value of the empty vector negative control. For peptide level responses determined by ELISA, if ≥2 replicate wells gave $OD_{450}$ values ≥0.15 and were ≥2-fold higher than negative-control values, responses were considered positive.

### Reporting summary

Further information on research design is available in the Nature Portfolio Reporting Summary linked to this article.

## Data availability

*TRB* CDR3 sequences are available at ImmuneAccess: https://clients. adaptivebiotech.com/pub/laing-2022-nc and Zenodo (https://doi.org/ 10.5281/zenodo.7141050). Epitopes reported in Supplementary materials were submitted to the Immune Epitope Database (IEDB) and include Accession numbers, with raw data for protein and peptide level screens deposited at Zenodo (https://doi.org/10.5281/zenodo. 7153680).

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

## Acknowledgements

The authors gratefully acknowledge the staff at the University of Washington Virology Research Clinic for their assistance with the clinical protocol, and the subjects for their participation. Tamana Khemai-Mehraban (HerpeslabNL) is acknowledged for her important contribution to in situ analysis of skin. Supported by NIH grant RO1AG064800 (D.M.K.) and NIH contracts HHSN272201400049C (D.M.K.) and 75N93019C00063 (D.M.K., G.M.G.M.V.) and stipends from the University of Washington Medical Student Research Training Program to SM and MEW.

## Author contributions

The study was devised, planned and managed by D.M.K., K.J.L., and G.M.G.M.V. K.J.L., W.J.D.O., G.M.G.M.V. and D.M.K. analyzed experimental data and wrote the manuscript. K.J.L. and V.L.C. directed T cell experimental procedures, assisted by C.L.M., S.M., M.E.W., M.K., R.T., N.N., and K.B. C.M. and D.S.S. performed antibody testing. G.M.G.M.V. and W.J.d.O. performed immunohistology. C.J. directed the clinical protocol. All authors reviewed and approved the manuscript.

## Competing interests

D.M.K. and K.J.L. are scientific advisors or consultants for Curevo and MaxHealth concerning candidate vaccines for herpes zoster. The findings and conclusions in this report are those of the authors and do not necessarily represent the official position of the Centers for Diseases Control and Prevention (CDC). The remaining authors declare no competing interests.
