## [Peer Review File · Nature Communications]

REVIEWER COMMENTS

Reviewer #1 (Remarks to the Author):

This study provides a thorough analysis of T cell numbers, subtypes and specificities over time in the skin of patients with recent zoster using same-patient controls (contralateral biopsies).

It is significant for the field in that it provides a comprehensive evaluation of Trm in herpes zoster, an area that has lagged well behind the large number of studies examining the role of Trm in HSV.

The work supports the conclusions of the authors. The stated hypothesis notwithstanding ("that a single, anatomically restricted, intense exposure to viral antigen leads to localized deposition of long-lasting virus-specific TRM in human skin"), this work is basically an examination of Trm in herpes zoster. The results mostly follow expectations and may not seem overly profound in and of themselves. However, the fact that this is a novel effort in herpes zoster as well as a very comprehensiveness study and methodologically strong study make it a solid contribution to the field that can be the foundation for future studies.

The study is small, so the statistically-based conclusions are somewhat limited. However, the core results of the study, even if viewed descriptively, enhance our understanding of T cell immunity in HZ.

The most important big-picture weakness of the paper is its very limited discussion of the clinical significance of this work. The authors touched only very briefly on the potential pathological/clinical significance of Trm in herpes zoster. Might the presence of Trm in herpes zoster simply be an immunologic artifact or does it serve any real purpose? The authors note that recurrences in the same dermatome are rare, but recurrence are uncommon in any case, so that is not too compelling. Are there other studies that could be done to evaluate the clinical relevance (or not) of Trm in herpes zoster? Another approach would be to add some additional thoughts of the relevance of this work beyond herpes zoster. One way or another, a bit more discussion on the clinical value of this work would be helpful.

A few other points to consider:

1. The patients enrolled in this study were clearly younger on the average, by about 15 years, than typical herpes zoster patients. Perhaps this should be discussed, and it may be worth noting if any differences were observed between the younger vs. older participants.

2. Had any of the patients received a shingles vaccine or, for that matter, a varicella vaccine? If so, do the authors have any thoughts on how this may or may not have affected the results?

3. Was any effort made to quantitate the severity of the individual subjects' herpes zoster? Might that have had an effect on the subsequent evaluations?

4. I assume that contralateral biopsies were taken from the same dermatome - is that correct?

Reviewer #2 (Remarks to the Author):

Laing et al examine the skin Varicella-zoster virus specific T cell response in healed Herpes zoster skin lesions over 45-360 days post onset as a unique example of the evolution, duration and function of human resident memory CD4 and CD8 T cell responses after a single Herpes zoster episode.

There are many potential cofounders, including persistence of viral antigens in the skin or reseeding from blood which they have addressed by parallel blood T cell studies and biopsying contralateral uninvolved skin as a control for healed rash biopsies. These are challenging studies both in consent from subjects for repeat biopsies to 360 days and also technically in documenting CD4 Na CD8 T cell responses in an HLA diverse population. It is not surprising that the number of biopsied subjects markedly diminished at later times. The authors have brought their collaborative repertoire of experience, skills and reagents, including a full complement of VZV ORFs, to provide a unique insight into resident memory T cell generation function and persistence from a single viral (re)challenge.

The figures are clear and innovative in presenting complex data with multiple host-viral parameters over time. Complimentary approaches such as figures 5A and B provide confidence in the specificity of results. The supplementary figures are useful in understanding the approaches and data in the main figures. Overall the data are very interesting and, for this type of investigation relatively consistent, however there are some contradictions and lack of clarity which is explored below.

Specific questions

1. How were biopsies and swabs for DNA accurately targeted to the healed rash, including at serial time points (ie were they adjacent, given the new clonotype recognising epitope ORF53 appearance at day 360 in Figure 5D)? Were photos of the rash available? Were studies of intra-rash variability conducted in the same subject? In this respect it is reassuring to see the consistency of ORF9, 18 and 62 recognising clonotypes across the serial time points in in Figure 5E. However, in figures 5D/E, supplementary figure 4B and supplementary Table 6 the recognition of ORF53 directly or as clonotypes appears to differ at days 45, 90 and 360, where it emerges from being 'subdominant'? Is there an explanation?

It was interesting to see the success they had in detecting VZV DNA from healed rashes. Were these just fragments or approximately full genome length?

2. Figure 2: In Results the authors claim that there is an increase in the proportion of regulatory T cells over time. Are the trends with FoxP3 over time in panel E significant? Please indicate the statistical tests used.

3. Figure 3: Panel B: the VZV specific T cells in the contralateral skin also seem to decline with time. Is this significant? If so, how do they explain this?

Panel C: the legend states n=1 but the figure shows 2 subjects. Why are there far fewer subjects for epitope breadth for CD4 T cell here than for CD8 T cells in Figure 4c.

The difference in numbers between panels B and C and at each time point could be confounders. Do the same trends with time occur when the specimens for each subject are linked?

Minor: In the legends to figures the abbreviations need to be spelled out at the end as usually done.

Figure 3A: for clarity the axes should be labelled 'Site' and 'Antigen' to avoid confusing CON and control!

Panel D and E legends are confused between those on the figures and in the main body of the text.

Point by point responses

We thank the reviewers for their supporting comments and suggestions on how to improve our manuscript. Our point by point responses are provided herein.

REVIEWER 1

1. Remarks to the Authors:

This study provides a thorough analysis of T cell numbers, subtypes and specificities over time in the skin of patients with recent zoster using same-patient controls (contralateral biopsies).

It is significant for the field in that it provides a comprehensive evaluation of Trm in herpes zoster, an area that has lagged well behind the large number of studies examining the role of Trm in HSV.

The work supports the conclusions of the authors. The stated hypothesis notwithstanding ("that a single, anatomically restricted, intense exposure to viral antigen leads to localized deposition of long-lasting virus-specific TRM in human skin"), this work is basically an examination of Trm in herpes zoster. The results mostly follow expectations and may not seem overly profound in and of themselves. However, the fact that this is a novel effort in herpes zoster as well as a very comprehensiveness study and methodologically strong study make it a solid contribution to the field that can be the foundation for future studies.

The study is small, so the statistically-based conclusions are somewhat limited. However, the core results of the study, even if viewed descriptively, enhance our understanding of T cell immunity in HZ.

Response:

No response required to these Remarks to the Authors.

2. Critiques embedded within Remarks to the Authors:

The most important big-picture weakness of the paper is its very limited discussion of the clinical significance of this work. The authors touched only very briefly on the potential pathological/clinical significance of Trm in herpes zoster. Might the presence of Trm in herpes zoster simply be an immunologic artifact or does it serve any real purpose? The authors note that recurrences in the same dermatome are rare, but recurrence are uncommon in any case, so that is not too compelling. Are there other studies that could be done to evaluate the clinical relevance (or not) of Trm in herpes zoster? Another approach would be to add some additional thoughts of the relevance of this work beyond herpes zoster. One way or another, a bit more discussion on the clinical value of this work would be helpful.

Response:

We agree with the Reviewer that more discussion of clinical significance would be a valuable contribution. To address this, we added the following paragraph to the Revised Discussion. The

text below contains PMID numbers, while the version in the Revised Discussion contains EndNote citations integrated with the rest of Revision.

The functional mechanisms of VZV-specific T cells at infected sites remain to be fully elucidated. Their potential to clear virus or promote tissue healing was not investigated in this report because we started biopsies after lesion resolution. However, based on data from active human HSV lesions {9525993, 11238653}, it is reasonable to propose that recruitment and/or *in situ* proliferation of VZV-specific CD4 and CD8 T cells occurs in skin during active HZ. Future research using HZ vesicle fluid or biopsy of active lesions combined with knowledge of VZV-reactive T cells determined via TCR sequencing or TCR databases {26810224, 27030598}, could facilitate further interrogation of the contribution made by infiltrating, resident, and recirculating {31278120} VZV-specific T cells to viral clearance, wound healing, and resolution of HZ lesions. The pro-inflammatory effects of VZV-specific T cells may contribute to lesion and HZ pain, severity and duration. Human autopsy studies reveal the trigeminal ganglia, a site of VZV latency and reactivation, contains T cells abutting neurons after the resolution of HZ supporting a potential for local antigen-reactive T cells to contribute to PHN pathogenesis {31572325}. VZV-specific T cells persisting after HZ resolution may also aid host defense against future episodes of HZ in the same dermatome, as seen in murine studies with HSV, whereby local skin CD8 T_{RM} provide anatomically-restricted protection against reinfection {19305395}. Unfortunately, current animal models of VZV infection {34524508} are unlikely to contribute to functional studies of local VZV-specific T_{RM}. Natural history investigations of local protection in human subjects against second episodes of HZ are also very challenging given the overall low incidence of recurrent HZ {31679866}. As a surrogate for HZ, the recently described model system in which inactivated VZV antigen is introduced into human skin {23284056} could be used to assess differences between healed HZ and control sites, with a focus on CD4 T cells. Use of replication-competent, attenuated VZV strain vOKA, licensed as a vaccine, might be a better future experimental medicine approach to study CD8 T_{RM} in human skin. It is not known if the highly clinically effective recombinant zoster vaccine increases skin-resident VZV-specific T cells. Such a finding would be consistent with, but not prove, a role for T_{RM} in the mechanism of action of this vaccine.

3. Critique

The patients enrolled in this study were clearly younger on the average, by about 15 years, than typical herpes zoster patients. Perhaps this should be discussed, and it may be worth noting if any differences were observed between the younger vs. older participants.

Response

We agree with the Reviewer that this is an interesting question but, unfortunately, we do not have sufficient subject numbers to accurately assess correlations between age and the immune variables we measured. We observed no correlation between age and HZ-site skin CD4 or CD8 T cell responses within our small cohort. Regarding age, we added to following text to the limitations section of the Revised Discussion: The text below contains a PMID number, while the version in the Revised Discussion contains EndNote citations integrated with the rest of Revision.

HZ is classically a disease of the aged and many of our subjects were relatively young. The median subject age of 47 is close the US-recommended age (50) for receiving the recombinant zoster vaccine. This cutoff was set based on the age of an upswing in HZ risk {30496358}. We observed no correlation between age and HZ-site skin CD4 or CD8 T cell responses within our

small cohort. Future studies could uncover potential differences in VZV-specific TRM with aging. Our case-finding method was unbiased regarding age. During recruitment for this study (2018-2020), uptake for shingles preventative vaccination was low in the US. Therefore, it is unlikely that our study was biased to younger persons related to low HZ rates in older persons.

4. Critique

Had any of the patients received a shingles vaccine or, for that matter, a varicella vaccine? If so, do the authors have any thoughts on how this may or may not have affected the results?

Response

Our clinical records indicate 3 subjects (81, 89, and 91) had self-reported receipt of one of the shingles vaccines. We have no data regarding varicella vaccination, but few of our subjects are likely to have received childhood varicella vaccine due to their age and the introduction of universal pediatric varicella vaccination in the US in 1995. We added the following statement to the Revised Results section under Subjects and Specimens:

Three subjects self-reported receipt of a zoster vaccine: subject 81 (recombinant zoster vaccine, 48 months prior to HZ), subject 89 (unknown type and time), subject 91 (recombinant zoster vaccine, unknown time). Varicella vaccination status was not captured but most subjects were born before universal pediatric varicella vaccination began in the US in 1995.

We have also added this sentence in the limitations paragraph in Discussion:

We note a few participants received shingles vaccination, but we are unable to assess whether this impacted the results.

5. Critique

Was any effort made to quantitate the severity of the individual subjects' herpes zoster? Might that have had an effect on the subsequent evaluations? I assume that contralateral biopsies were taken from the same dermatome - is that correct?

Response

We greatly appreciate this comment for the Reviewer. We did not capture the severity of herpes zoster in this study but will certainly consider assessing this in future studies. We agree this may have an association with persistent local immune responses, the focus of this study.

The Reviewer is correct regarding the contralateral biopsies. All contralateral biopsies were taken from the same dermatome as the HZ site biopsies. As mentioned in a response to Reviewer #2, we obtained photos of the original rash and obtained biopsies from the rash site and the contralateral site for all participants to document the precise locations. We Revised the sentence in the Subjects and Specimens section of the Methods that mentioned HZ site biopsy location:

Biopsies were performed at subject-described areas of prior HZ or the contralateral side within the same dermatome.

REVIEWER 2

Remarks to the Authors:

Laing et al examine the skin Varicella-zoster virus specific T cell response in healed Herpes zoster skin lesions over 45-360 days post onset as a unique example of the evolution, duration and function of human resident memory CD4 and CD8 T cell responses after a single Herpes zoster episode.

There are many potential cofounders, including persistence of viral antigens in the skin or reseeded from blood which they have addressed by parallel blood T cell studies and biopsying contralateral uninvolved skin as a control for healed rash biopsies. These are challenging studies both in consent from subjects for repeat biopsies to 360 days and also technically in documenting CD4 Na CD8 T cell responses in an HLA diverse population. It is not surprising that the number of biopsied subjects markedly diminished at later times. The authors have brought their collaborative repertoire of experience, skills and reagents, including a full complement of VZV ORFs, to provide a unique insight into resident memory T cell generation function and persistence from a single viral (re)challenge.

The figures are clear and innovative in presenting complex data with multiple host-viral parameters over time. Complimentary approaches such as figures 5A and B provide confidence in the specificity of results. The supplementary figures are useful in understanding the approaches and data in the main figures. Overall the data are very interesting and, for this type of investigation relatively consistent, however there are some contradictions and lack of clarity which is explored below.

Response

Remarks to the Authors do not contain any critiques

Specific questions:

1. Critique

How were biopsies and swabs for DNA accurately targeted to the healed rash, including at serial time points (ie were they adjacent, given the new clonotype recognising epitope ORF53 appearance at day 360 in Figure 5D)? Were photos of the rash available?

Response

Pre and post biopsy photos were taken for each biopsy. The site(s) of sequential biopsies were approximated as closely as possible to the site(s) of previous biopsies. There are often areas of pigmentation change after biopsy that allows for adjacent biopsies to be accurately placed. To make this clearer in Revision, we added the following statement to the Revised Methods section:

Pre- and post-biopsy photographs and areas of post-biopsy hypo- or hyper-pigmentation, if present, were used to guide the locations of sequential biopsies as closely together as possible.

2. Critique

Were studies of intra-rash variability conducted in the same subject?

Response

We understand (and agree with) Reviewer 2 that some variability would be anticipated in different biopsies from the same site. We mention this in our limitations paragraph in the original manuscript. From our experience with other studies, we predict that rare clonotype specificities may differ between adjacent sites, while dominant clonotypes and clonotype breadth will remain consistent. Although, in this study, we did not routinely compare T cell responses in adjacent biopsies from the same timepoint within subject using the same method, the data presented in Fig. 5 and Supplementary Fig. 4 confirm we detected T cells with the same epitope specificity in adjacent biopsies using completely different methods (functional screens of T cell line screens using APC expressing VZV genes, *in situ* tetramer staining, and TCR sequencing). We did not Revise the manuscript text in response to this comment.

3. Critique

In this respect it is reassuring to see the consistency of ORF9, 18 and 62 recognising clonotypes across the serial time points in in Figure 5E. However, in figures 5D/E, supplementary figure 4B and supplementary Table 6 the recognition of ORF53 directly or as clonotypes appears to differ at days 45, 90 and 360, where it emerges from being 'subdominant'? Is there an explanation?

Response

Thank you for catching this omission. Supplementary Fig. 4B was missing information that ORF53 and ORF62 peptides were tested on the day 360 T cell line (matching the tetramer data in Fig. 5C). To address this, in the Revision, we have changed Supplementary Fig. 4B to add timepoint details for each peptide in the key for the graph concerned. The data in Supplementary Fig. 4B agree with Fig 5D/E and with the data in Supplementary Table 6 and are based off these data.

4. Critique

It was interesting to see the success they had in detecting VZV DNA from healed rashes. Were these just fragments or approximately full genome length?

Response

These were short regions of genomic viral DNA. The PCR methods used are referenced in the original submission and no changes are made in the Revision.

5. Critique

Figure 2: In Results the authors claim that there is an increase in the proportion of regulatory T cells over time. Are the trends with FoxP3 over time in panel E significant? Please indicate the statistical tests used.

Response

The Reviewer is correct that we did not observe a statistically significant change in FoxP3 over time. While we did not claim to see trends in FoxP3 in the Results in the original submission, we removed the statements concerning Tregs from the second paragraph of the Revised Discussion. The statistical test used was Wilcoxon and is now explained more adequately in the revised Methods Statistics section and Figure legend.

6. Critique

Figure 3: Panel B: the VZV specific T cells in the contralateral skin also seem to decline with time. Is this significant? If so, how do they explain this?

Response

We agree it is interesting that VZV-specific CD4 T cells are observed on the contralateral side at day 45 and appear to decrease by day 90. However, there is no statistically significant decrease in the contralateral CD4 T cell response in Fig. 3B between days 45 and 90 when using a Wilcoxon test. We note that the median response on day 45 was really quite low, about 0.03% of CD4 T cells with reactivity to whole virus, which in our experience barely “on screen” for this type of assay. We have not changed the revised manuscript regarding this point.

7. Critique

Panel C: the legend states n=1 but the figure shows 2 subjects. Why are there far fewer subjects for epitope breadth for CD4 T cell here than for CD8 T cells in Figure 4c.

Response:

Thank you for catching this typing error. This has been corrected to n = 2 in the figure legend. We surveyed fewer samples for VZV-protein specific CD4 T cell responses due to limitations in sensitivity of the CD4 T cell assay, which we also detail in the next critique. We only completed proteome-wide ORF screens (Fig. 3C) in persons with reactivity to whole VZV (Fig. 3B) above the 3% threshold specified in Methods in the original submission.

8. Critique

The difference in numbers between panels B and C and at each time point could be confounders. Do the same trends with time occur when the specimens for each subject are linked?

Response

The Reviewer is correct that a different number of subjects are represented in Fig. 3B versus Fig. 3C, and also at different time points within Fig 3B and within Fig. 3C. We only completed proteome-wide ORF screens (Fig. 3C) in persons with reactivity to whole VZV (Fig. 3B) above the 3% threshold specified in Methods in the original submission. The proportions of persons reaching threshold at each time point and anatomic location for biopsy are shown in Supp Fig 1 A. There are fewer people at later time points within Fig. 3B and Fig. 3C because some participants dropped out. With regards to the issue of confounding, it is possible that subject dropout leading to a lower number of observations at the later time points might make our

conclusions less certain by day 360. We acknowledge this in the Revised Discussion with the modified sentence:

Some subjects declined the final biopsy, leading to a lower number of observations at the later time points, which might make our conclusions less certain by day 360.

With regards to trends with time, we clarify here that Fig. 3B does not statistically analyze within-subject changes over time. Rather, Fig. 3B notes that at each time point, higher CD4 T cell responses are noted at the HZ site than at the contralateral site. The statistical test is a paired test, conforming to Reviewer's suggested linked test. In the Statistics section of Methods in the Revised manuscript, we clarify that a Wilcoxon paired test was used to compare HZ to con within time-point. This test was mentioned in the Figure 3 legend in the original manuscript.

9. Critique

Minor: In the legends to figures the abbreviations need to be spelled out at the end as usually done.

Response

Abbreviations were spelled out where appropriate in the figure or figure legend, according to journal guidelines.

10. Critique

Figure 3A: for clarity the axes should be labelled 'Site' and 'Antigen' to avoid confusing CON and control!

Response

We made this correction to Figure 3.

11. Critique

Panel D and E legends are confused between those on the figures and in the main body of the text.

Response

We made this correction to the Figure legend.

REVIEWERS' COMMENTS

Reviewer #1 (Remarks to the Author):

I would like to thank the authors for their thoughtful responses to my comments. All of my concerns have been adequately addressed.

Reviewer #2 (Remarks to the Author):

My comments and criticisms have been satisfactorily addressed